# KnobGen: Controlling the Sophistication of Artwork in Sketch-Based Diffusion Models

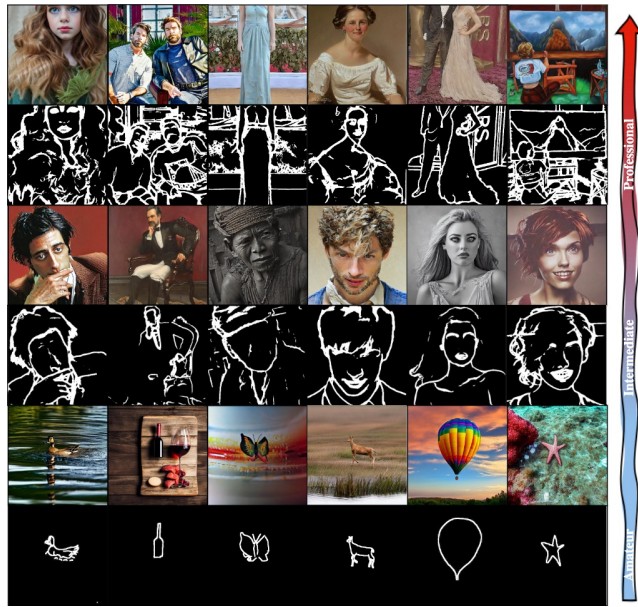

Figure 1. **KnobGen**. Our method democratizes sketch-based image generation by effectively handling a broad spectrum of sketch complexity and user drawing ability—from novice sketches to those made by seasoned artists—while maintaining the natural appearance of the image.

## Abstract

*Recent advances in diffusion models have significantly improved text-to-image (T2I) generation, but they often struggle to balance fine-grained precision with high-level control. Methods like ControlNet and T2I-Adapter excel at following sketches by seasoned artists but tend to be replicating unintentional flaws in sketches from novice users. Meanwhile, coarse-grained methods, such as sketch-based abstraction frameworks, offer more accessible input handling but lack the precise control needed for professional use. To address these limitations, we propose **KnobGen**, a dual-pathway framework that democratizes sketch-based image generation by adapting to varying levels of sketch complexity and user skill. KnobGen uses a Coarse-Grained Controller (CGC) module for high-level semantics and a Fine-Grained Controller (FGC) module for detailed refinement. The relative strength of these two modules can be adjusted through our **knob** inference mechanism to align with the user's specific needs. These mechanisms ensure that KnobGen can flexibly generate images from both novice sketches and those drawn by seasoned artists. This maintains control over the final output while preserving the natural appearance of the image, as evidenced on the MultiGen-20M dataset and a newly collected sketch dataset.*

## 1. Introduction

Diffusion models (DMs) have revolutionized text-to-image (T2I) generation by generating visually rich images based on text prompts, excelling at capturing various levels of detail—from textures to high-level semantics [25, 26, 34, 36, 38]. Despite their success, one of the primary limitations of these models is their inability to precisely convey spatial layout of the user-provided sketches. While text prompts can describe scenes, they struggle to capture complex spatial features, which makes it challenging to align generated images with user intent. This is particularly intensified when these users vary in skill and experience [3, 13, 43, 52].

To improve spatial control, sketch-conditioned DMs like ControlNet [54], T2I-Adapter [24], and ControlNet++ [19] have introduced mechanisms to allow users to input sketches that guide the generated image. However, these approaches primarily cater to artistic sketches with intricate details, which poses a challenge for novice users. When presented with rough sketches, these models rigidly align to unintentional flaws, producing results that misinterpret the user's intent and fail to achieve the desired visual outcome. Furthermore, as shown in Figure 2 we observed that the quality and alignment of the generated images with the input sketch are highly sensitive to the weighting parameter that governs the model's dependence on the condition.

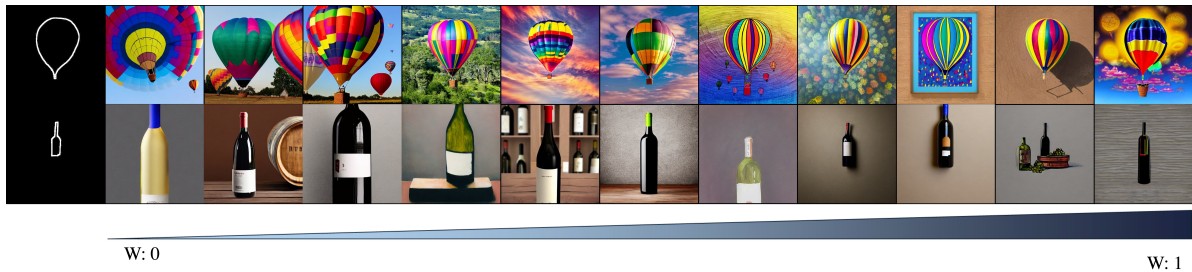

W: 0

W: 1

Figure 2. **Qualitative results demonstrating the impact of varying the weighting scheme in T2I-Adapter model**. Lower weights result in images that poorly align with the input sketch in terms of spatial conformity, while higher weights improve spatial conformity of the generated image to the input sketch. However, higher weight compromises the natural appearance of the generated images.

In contrast, some frameworks such as [18, 47][1] have attempted to address the needs of novice users by introducing sketch abstraction. Although this democratizes the generation process, [18] is limited to covering only 125 categories of sketch subjects and cannot handle unseen categories, significantly limiting the generalizability of the pre-trained DM to a limited number of subjects. Moreover, its abstraction-aware framework is not suitable for artistic-level sketches whose purpose is to guide the DM to follow a particular spatial layout. Additionally, the removal of the text-based conditioning in DM makes these models ignore the semantic power provided by text in diffusion models trained on large-scale image-text pairs. Additionally, it limits their ability to differentiate between visually similar but semantically distinct objects- such as zebra and horse. In [47] the latent space of DM is rigidly aligned with that of the sketch, resulting in maximal reliance on the input sketch.

In a nutshell, existing methods for sketch-based image generation tend to focus on either end of the user-level spectrum. As illustrated in Figure 3.a ,fine-grained conditioning modules like ControlNet and T2I-Adapter are designed to handle only artistic-grade sketches, while amateur-oriented approaches [18] in Figure 3.b cater to novice sketches without text guidance. These methods often fail to integrate both fine-grained and coarse-grained control across different user types and sketch complexities.

To address these challenges, we propose **KnobGen**, a dual-pathway framework designed to empower a pre-trained DM with the capability to handle both professional and amateur-oriented approaches. KnobGen seamlessly integrates fine-grained and coarse-grained sketch control into a unified architecture, allowing it to adapt to varying levels of sketch complexity and user expertise. Our model is built on two key pathways, *Macro Pathway* and *Micro Pathway*. The Macro Pathway extracts the high-level visual and language semantics from the sketch image and the text prompt using CLIP encoders and injects them into the

DM via our proposed **Coarse-Grained Controller** (CGC). The Micro Pathway injects low-level features directly from sketch through our **Fine-Grained Controller** (FGC).

Additionally, we propose two new approaches for training and inference in order to maintain a robust control of the Micro and Macro Pathways in the conditional generation. First, we introduce **Modulator**, a mechanism dynamically adjusting the influence of the FGC during training, ensuring that the CGC dominates in the early training phase to prevent overfitting to low-level sketch features extracted by the FGC module. This allows the model to optimally rely on both Pathways to capture high- and low-level spatial and semantic features. At inference, the **Knob** mechanism offers user-driven control during denoising steps, allowing adjustment of the level of fidelity between the generated image and the user's inputs- sketch and text- by manipulating Micro and Macro Pathways. These new training and inference approaches ensure that KnobGen effectively handles not only novice sketches but also artistic-grade ones. Our key contributions are as follows:

- **Dual-Pathway Framework for Adaptive Sketch-Based Image Generation**: KnobGen introduces a novel dual-pathway design that balances fine-grained and coarse-grained pathways, providing controlled flexibility for sketches with varied levels of details. This integration extends KnobGen's applicability across diverse user types, from novice sketchers to seasoned artists, addressing a major gap in prior sketch-guided DM.
- **Dynamic Modulator to Harmonize Coarse and Fine Detail During Training**: Our *modulator* mechanism tunes the influence of coarse and fine-grained pathways throughout training, overcoming the tendency of fine-grained details to dominate early stages. By balancing these inputs, our approach achieves optimal spatial layout and feature refinement, which SOTA models lack due to their reliance on fixed weighting schemes.
- **Inference-Time Knob for User-Controlled Sketch Fidelity and Realism**: Unlike uniform weighting schemes in other models, our *knob* mechanism variably introduces

---

[1]The codes and model weights at the time of submission were unavailable which prevents reproducibility.

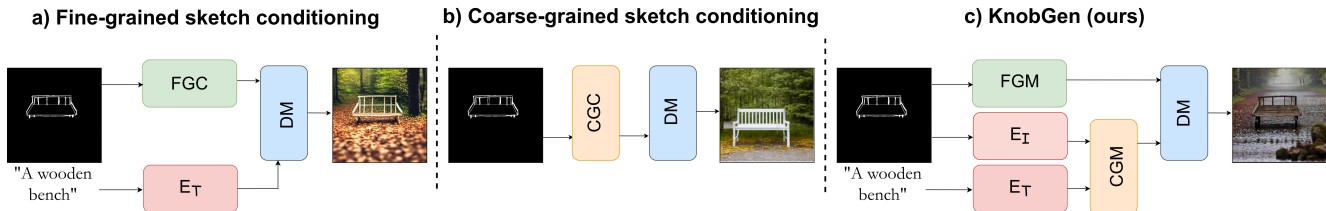

**a) Fine-grained sketch conditioning**    **b) Coarse-grained sketch conditioning**    **c) KnobGen (ours)**

Figure 3. **Comparison across various sketch-control in DM.** (a) fine-grained control based method such as ControlNet or T2I-adapter rigidly resembles a novice sketch resulting in an unrealistic image (b) abstraction-aware frameworks such as [18] fails to capture fine grained-detials without text guidance(c) while our proposed KnobGen smoothes out the imperfection of the user drawing and preserves the features of the novice sketch. FGC: Fine-grained Controller, CGC: Coarse-grained Controller, $E_T$: Text Encoder, $E_I$: Image Encoder, DM: Diffusion Model.

detail based on user input, preserving both spatial adherence to sketches and natural image appearance. This user-driven control allows KnobGen to flexibly adapt to diverse user needs.

## 2. Related Work

### 2.1. Diffusion Models

Recent advances in DM have enabled high-quality image generation with improved sample diversity [4, 10, 11, 14, 23, 26, 28, 29, 35, 38, 42], often exceeding the performance of Generative Adversarial Networks (GAN) [7, 15, 16, 40]. DMs are built on the concept of diffusion processes, where data are progressively corrupted by noise over several timesteps. The models learn to reverse this process by iteratively denoising noisy samples, transforming pure noise back into the original data distribution. Several studies, such as DDIM [44], DPM-solver [20], and Progressive Distillation [39], have focused on accelerating DMs' generation process through more efficient sampling methodologies. To address the high computational costs of training and sampling, recent research has successfully employed strategies to project the original data into a lower-dimensional manifold, with DMs being trained within this latent space. Representative methods include LSGM [46], LDM [36], and DALLE-2 [35] which leverage latent space.

### 2.2. Text-to-Image Diffusion

In addition to producing high-quality and diverse samples, DMs offer superior controllability, especially when guided by textual prompts [2, 5, 30, 36, 41, 50]. Imagen [38] employs a pretrained large language model (e.g., T5 [33]) and a cascade architecture to achieve high-resolution, photorealistic image generation. LDM [36], also known as Stable Diffusion (SD), performs the diffusion process in the latent space with textual information injected into the underlying UNet through a cross-attention mechanism, allowing for reduced computational complexity and improved generation fidelity. To further address challenges when handling complex text prompts with multiple objects and object-attribution bindings, RPG [53] proposed a training-free framework that harnesses the chain-of-thought reasoning capabilities of multimodal large language models (LLMs) to enhance the compositionality of T2I generation. Ranni [6] tackles this problem by introducing a semantic panel that serves as an intermediary between text prompts and images; an LLM is finetuned to generate semantic panels from text which are then embedded and injected into the DM for direct composition. Our proposed method aligns with the SD paradigm but diverges by incorporating a composite module that combines textual information with coarse-grained information from sketch inputs, thereby injecting more comprehensive high-level semantics into the diffusion model.

### 2.3. Conditional Diffusion with Semantic Maps

As textual prompts often lack the ability to convey detailed information, recent research has explored conditioning DMs on more complex or fine-grained semantic maps, such as sketches, depth maps, normal maps, etc. Works such as T2I-Adapter [24], ControlNet [54], and SCEdit [13], leverage pretrained T2I models but employ different mechanisms to interpret and integrate these detailed conditions into the diffusion process. UniControl [31] proposes a task-aware module to unify $N$ different conditions (i.e. $N = 9$) in a single network, achieving promising multi-condition generation with significantly fewer model parameters compared to a multi-ControlNet approach. While [18] attempts to democratize sketch-based diffusion models, their approach faces several significant limitations, as discussed in the Introduction section. In contrast, our dual-pathway method integrates both fine-grained and coarse-grained sketch conditions while maintaining the option for textual prompts.

## 3. Method

The design of KnobGen ensures that low- and high-level details from the conditional signal, i.e. the sketch, are incorporated in a balanced manner, both during *training* and

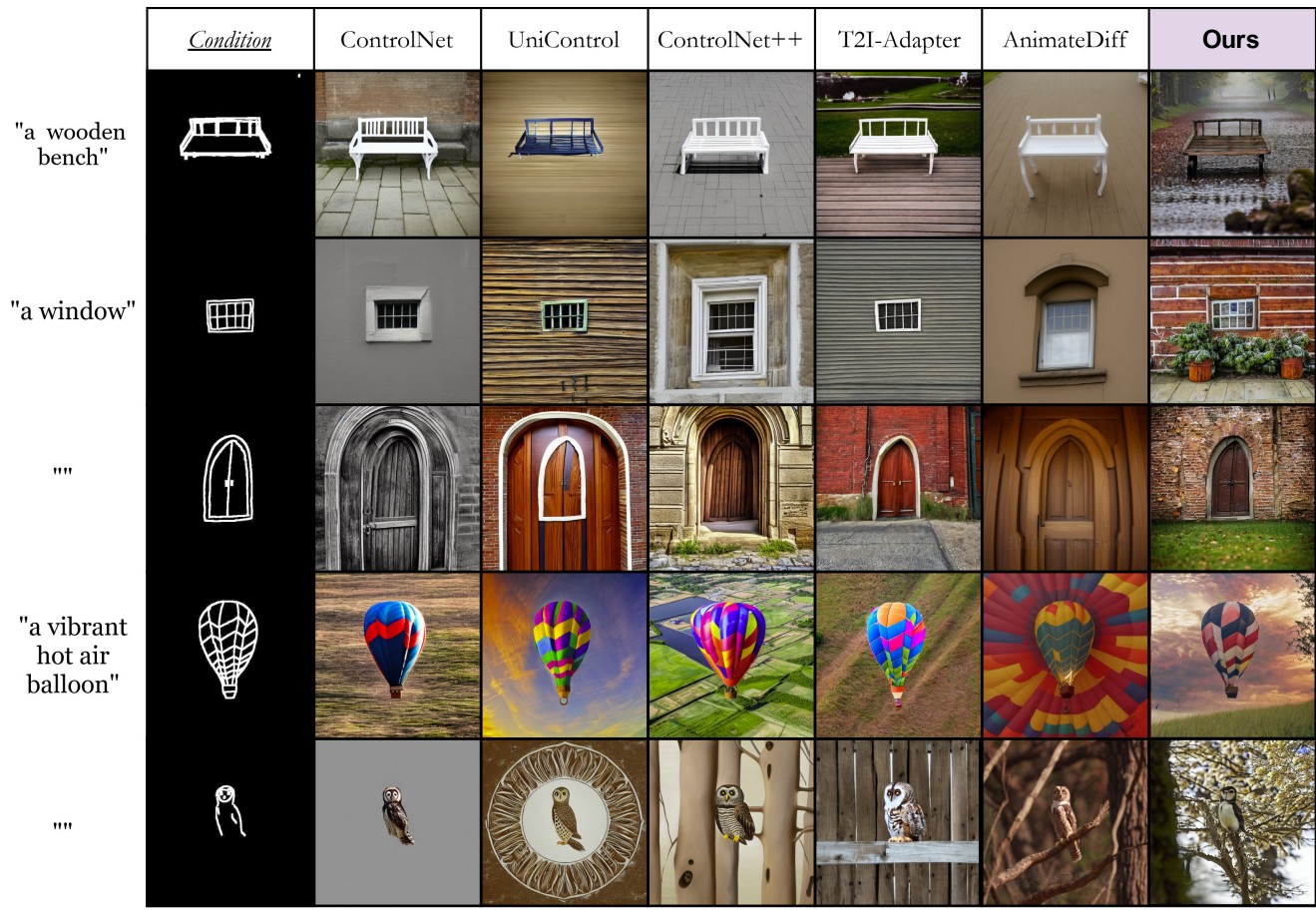

| | *Condition* | ControlNet | UniControl | ControlNet++ | T2I-Adapter | AnimateDiff | **Ours** |
|---|---|---|---|---|---|---|---|
| "a wooden bench" | | | | | | | |
| "a window" | | | | | | | |
| "" | | | | | | | |
| "a vibrant hot air balloon" | | | | | | | |
| "" | | | | | | | |

Figure 4. **KnobGen vs. baseline on novice sketches**. KnobGen handles novice sketches by injecting features from the Micro and Macro Pathways in a controlled manner. Dual pathway design ensures that the generated image is faithful to the spatial layout of the original input sketch and the image has a natural appearance. Baseline methods, however, exhibit difficulty in maintaining these desired properties in their generations. We also provide examples with null prompt as an ablation study to demonstrate the robustness of KnobGen.

*inference*. In section 3.3, we introduce the modulator, illustrated in Figure 5.A, which harmonizes the influence of fine-grained control in the training phase. The modulator prevents the fine-grained control from overpowering the coarse-grained control signal in the early training stages—a common challenge in generative models [51] that SOTA diffusion models often overlook. In section 3.4, we further describe our knob mechanism, Figure 5.C, which adaptively adjusts the level of detail during inference to align with the user's skill level.

### 3.1. Preliminary

**Stable Diffusion** Diffusion models [11] define a generative process by gradually adding noise to input data $z_0$ through a Markovian forward diffusion process $q(z_t|z_0)$. At each timestep $t$, noise is introduced into the data as follows:

$$z_t = \sqrt{\bar{\alpha}_t}z_0 + \sqrt{1 - \bar{\alpha}_t}\epsilon, \quad \epsilon \sim \mathcal{N}(\mathbf{0}, \mathbf{I}), \quad (1)$$

where $\epsilon$ is sampled from a standard Gaussian distribution, and $\bar{\alpha}_t = \prod_{s=0}^{t} \alpha_s$, with $\alpha_t = 1 - \beta_t$ representing a differentiable function of the timestep $t$. The diffusion process converts $z_0$ into pure Gaussian noise $z_T$ over time.

The training objective for diffusion models is to learn a denoising network $\epsilon_\theta$ that predicts the added noise $\epsilon$ at each timestep $t$. The loss function, commonly referred to as the denoising score matching objective, is expressed as:

$$\mathcal{L}(\epsilon_\theta) = \sum_{t=1}^{T} \mathbb{E}_{z_0 \sim q(z_0), \epsilon \sim \mathcal{N}(\mathbf{0}, \mathbf{I})} \left[ \| \epsilon_\theta(\sqrt{\bar{\alpha}_t}z_0 \right. $$
$$\left. + \sqrt{1 - \bar{\alpha}_t}\epsilon) - \epsilon \|_2^2 \right]. \quad (2)$$

In controllable generation tasks [24, 54], where both image condition $c_v$ and text prompt $c_t$ are provided, the diffusion loss function can be extended to include these conditioning inputs. The loss at timestep $t$ is modified as:

$$\mathcal{L}_{\text{train}} = \mathbb{E}_{z_0,t,c_t,c_v,\epsilon \sim \mathcal{N}(0,1)} \left[ \|\epsilon_\theta(z_t, t, c_t, c_v) - \epsilon\|_2^2 \right], \quad (3)$$

where $c_v$ and $c_t$ represent the visual and textual conditioning inputs, respectively.

During inference, given an initial noise vector $z_T \sim \mathcal{N}(\mathbf{0}, \mathbf{I})$, the final image $x_0$ is recovered through a step-by-step denoising process [11], where the denoised estimate at each step $t$ is calculated as:

$$z_{t-1} = \frac{1}{\sqrt{\alpha_t}} \left( z_t - \frac{1 - \alpha_t}{\sqrt{1 - \bar{\alpha}_t}} \epsilon_\theta(z_t, t, c_t, c_v) \right) + \sigma_t \epsilon, \quad (4)$$

with $\epsilon_\theta$ being the noise predicted by the U-Net [37] at timestep $t$, and $\sigma_t = \frac{1 - \bar{\alpha}_{t-1}}{1 - \bar{\alpha}_t} \beta_t$ representing the variance of the posterior Gaussian distribution $p_\theta(z_0)$. This iterative process gradually refines $z_t$ until it converges to the denoised image $z_0$.

### 3.2. Dual Pathway

Figure 5 demonstrates our model, a dual-pathway framework that harmonizes high-level semantic abstraction with precise, low-level control over visual details. The integration of the **CGC** module and **FGC** module enables Knob-Gen to adaptively inject high-level semantics and low-level features throughout the denoising process. This design ensures that the model can scale its output complexity based on user input, thus supporting a wide spectrum of sketch sophistication levels.

#### 3.2.1. Macro Pathway

Diffusion models typically rely on text-based conditioning using CLIP text encoders [32] to capture high-level semantics [26, 34, 38], but this approach often misses out on structural cues inherent to other modalities, such as sketches. Although models such as CLIP [32] encode visual features and textual semantics, they remain biased toward coarse-grained features [1, 48]. In our **CGC** module, Figure 5.B, we used this fact to our advantage to fuse a high-level visual and linguistic understanding to control DM generation by incorporating both text and image embeddings through a cross-attention mechanisms.

**Coarse-grained Controller (CGC):** In our CGC module, we leverage the trained CLIP text encoder and its corresponding image encoder variant available in the pretrained Stable Diffusion Model [36]. Our CGC module first takes the raw sketch image (condition) and prompt as input. Using the CLIP image and text encoders, the CGC module first projects them into $x_i \in \mathbb{R}^{256 \times 1024}$ and $x_p \in \mathbb{R}^{77 \times 768}$ which are the image and text embeddings. A cross-attention mechanism then fuses these embeddings to produce a multimodal

representation that combines textual semantics and visual cues. This enables the diffusion process to encode the high-level semantics from text while explicitly integrating spatial features from the sketch using the Clip image encoder. The cross-attended embeddings are injected into layers of the denoising U-Net to preserve the coarse-grained visual-textual features throughout the diffusion process. Detailed discussion of the CGC module is in the Appendix 7.

#### 3.2.2. Micro Pathway

For artistic users, preserving fine-grained details such as object boundaries and textures is essential. The **Fine-Grained Controller (FGC)** is designed to address these requirements by integrating pretrained modules such as Control-Net [54] and the T2I-Adapter [24], which excel in capturing these intricate features. Our Micro Pathway can utilize any pretrained fine-grained controller module which shows the flexibility of our proposed framework.

Incorporating these modules into our micro pathway allows the model to capture detailed, sketch-based features at multiple denoising stages. This pathway complements the coarse-grained features extracted by the CGC module, ensuring that the model not only preserves high-level semantic coherence, but also maintains visual fidelity and spatial accuracy with respect to sketch. Additionally, the FGC module ensures that the model handles professional-grade sketches with precision.

### 3.3. Modulator at training

One of the key innovations in KnobGen is the *tanh-based* modulator, which regulates the contributions of the micro and macro pathways during training, Figure 5.A. Based on our experiments in section 4.4, the incorporation of micro pathway in the early epochs of training process overshadows the effect of our macro pathway. Not only does this phenomenon lead to a model that overfits low-level features of the sketch, but it also prevents the model from generalizing to broader spatial and conceptual features. To mitigate this, we employ a modulator that progressively increases the impact of the Micro Pathway, i.e. the FGC module, during training. The modulator is based on a smooth tanh function:

$$m_t = m_{\min} + \frac{1}{2} \left( 1 + \tanh(\underbrace{k \cdot \frac{t}{T} - 3}_{\psi}) \right) \cdot (m_{\max} - m_{\min})$$

$$(5)$$

Here, $t$ is the current epoch, $T$ is the total number of epochs of training, $k = 6$, $\psi \in [-3, 3]$, $m_{\min} = 0.2$ and $m_{\max} = 1$ where $m_{\min}$ and $m_{\max}$ define the range within which the modulator effect (in percent), i.e. $m_t$, will vary over the course of the epochs. In order to choose $m_{\min}$, we heuristically found that the maximum lower bound for neg-

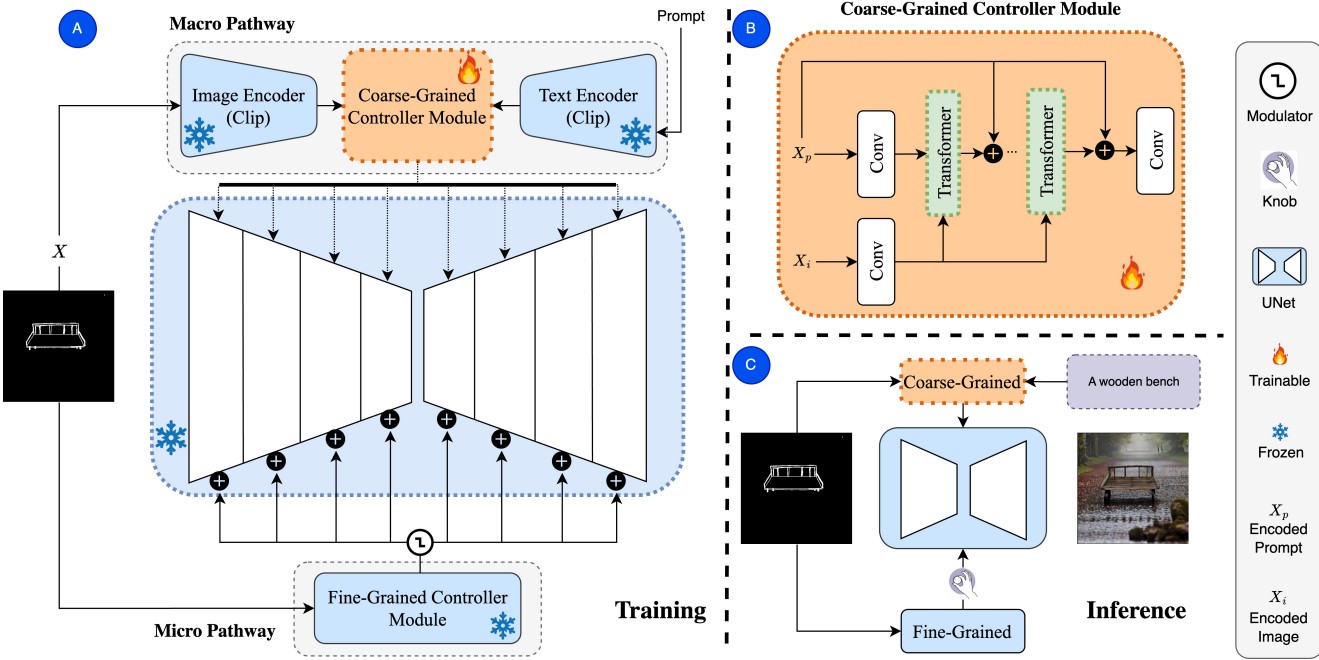

Figure 5. **Overview of KnobGen during training and inference**. A illustrates the training process, where the CGC and FGC modules are dynamically balanced by the modulator. B expands on the CGC module, detailing how high-level semantics from both text and image inputs are integrated. C shows the inference process, including the knob mechanism that allows user-driven control over the level of fine-grained detail in the final image.

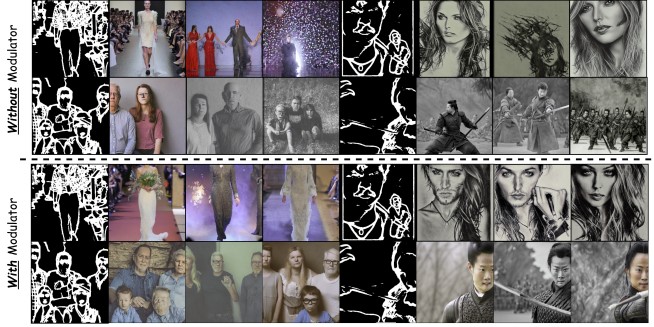

Figure 6. **Comparative results showcasing the impact of the Modulator in the training process**. The top side of the figure displays results generated by the model trained without the Modulator, while the bottom part illustrates outputs from the model trained with the Modulator.

ligible effect of the FGC is at $m_{\min} = 0.2$. We did not conduct an extensive hyperparameter search for $m_{\min}$ and only chose this value based on our observation of different case studies. As seen in Figure 5.A, the *module* ensures that diffusion is more affected by the Macro Pathway and less by the Micro Pathway in the early stages of training. As the training progresses, $m_t$ for the Micro Pathway approaches 1 and as a result our FGC module will have an equal impact in the training as that of the CGC. By gradually modulat-

ing the influence of the Micro Pathway, we prevent the premature weakening of high-level features presented by the Macro Pathway, and ensure that both pathways contribute optimally throughout the training. The effectiveness of our modulator is experimented in section 4.4.

**Remark**.We selected the tanh function for its gradual transition across epochs which enables balanced modulation of coarse and fine-grained contributions during training. While we did not test other functions, the tanh function's properties effectively support stable learning.

## 3.4. Inference Knob

In typical diffusion models, the early denoising steps during inference focus on generating high-level spatial features, while the later steps refine finer details [11, 21]. In our dual-pathway model, this mechanism is explicitly implemented by our proposed *inference-time Knob*. This is essentially a user-controlled tool (Figure 5.C) that determines the range of how much abstraction or rigid alignment with respect to the input sketch is desired by the user.

We introduce $\gamma$ variable as our **Knob** parameter. Let the total number of denoising steps be $S$, and $\gamma$ represent the step at which fine-grained details cease to influence the denoising process. The inference knob influnce the impact of the CGC and FGC modules at inference-time, allowing users to adjust $\gamma$ depending on their desired level of detail:

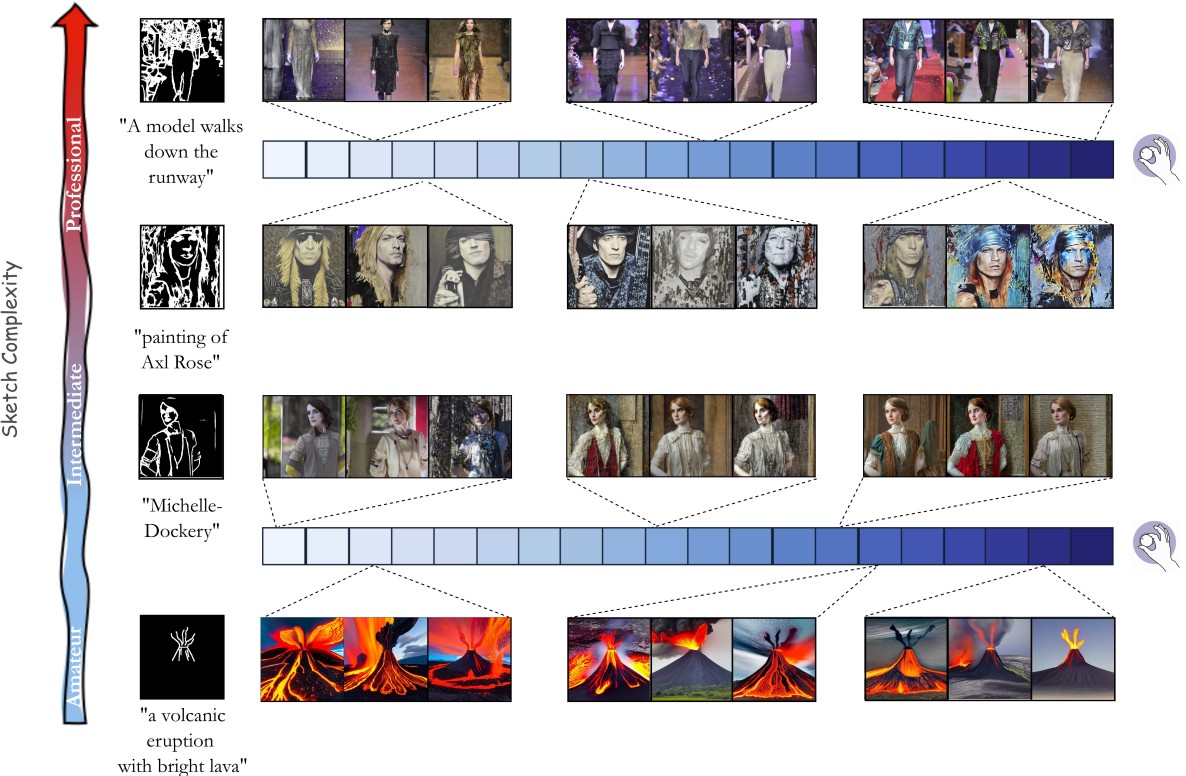

Figure 7. **Impact of the knob mechanism across varying sketch complexities**. From top to bottom, the sketches increase in complexity. The horizontal color spectrum represents the knob values, with light blue on the left ($\gamma$=20) indicating minimal reliance on the sketch, and dark blue on the right ($\gamma$=50) representing maximal reliance.

$$f_\ell(t) = \begin{cases} f_{\text{coarse}}(t) + f_{\text{fine}}(t), & \text{if } t \leq \gamma, \\ f_{\text{coarse}}(t), & \text{if } t > \gamma, \end{cases} \quad \begin{array}{c} \forall\, \ell \in \\ \{\text{U-Net layers}\} \end{array}$$

In this equation, $t$ represents the current denoising step during the inference. The parameter $\gamma$ acts as the knob value, determining the threshold at which the injection of fine-grained features ceases. When the denoising step $t$ is less than or equal to $\gamma$, both coarse-grained features $f_{\text{coarse}}(t)$ and fine-grained features $f_{\text{fine}}(t)$, generated by the macro and micro pathways respectively, are injected into the U-Net across layers, denoted by $\ell$. However, when $t$ exceeds $\gamma$, only the coarse-grained features $f_{\text{coarse}}(t)$ are injected into the U-Net.

A lower $\gamma$ value results in more abstract outputs with respect to the original input sketch, while a higher value makes the model produce images that closely match the sketch's finer details. This adaptive control allows Knob-Gen to accommodate a wide range of user preferences and input complexities shown in Figure 7, ensuring that both novice and artists can generate images that align with their expectations. The effectiveness of our proposed Knob mechanism is illustrated in Appendix( 9.2).

**Remark**. Unlike the T2I-Adapter [24] weighting approach, which applies a *uniform* weight over the entire denoising process (see Figure 2), our knob mechanism introduces a flexible adjustment. This mechanism allows users to selectively balance fine-grained and coarse-grained details throughout denoising, tailoring image generation to the preferred level of detail, as demonstrated in Figure 7.

## 4. Experiment

We conducted several qualitative and quantitative experiments to validate the effectiveness of KnobGen. The qualitative experiments showcase the effectiveness of our approach in guiding the DM based across different sketch complexities. The qualitative experiments evaluate our model against widely-used baselines on different generation metrics such as CLIP and FID scores. We used pretrained ControlNet and T2I-Adapter as our FGC module throught all our experimentation. According to the parameters defined in section 3.4, $\gamma = 20$ and $S = 50$. These values were heuristically selected and were used consistently in all experiments and baselines.

The extension of the qualitative experiments is available in the Appendix (9). More quantitative and user study result

are in Appendix (8). Furthermore, details about the setup used in the training and evaluation are in the Appendix (6).

## 4.1. Qualitative Results

Our qualitative results demonstrate the flexibility and effectiveness of KnobGen in handling varying sketch qualities. KnobGen is able to seamlessly adapt to sketches from rough amateur drawings to refined professional ones, highlighting its ability to cover the entire spectrum of user expertise. Figure 7 illustrates the impact of our knob mechanism, where increasing the knob value (left to right) progressively improves the fidelity to the sketch input. This dynamic adjustment enables precise control over the level of detail, allowing users to fine-tune generation outputs. More qualitative results with different input conditions and modes, such as no prompt, professional sketch and free-chyle sketch are provided in the Appendix (9.2).

## 4.2. Comparison vs. baselines

In order to conduct a fair comparative study, we evaluated KnobGen against baselines such as [19, 24, 54] on professional-grade sketches, novice ones and a spectrum in between. Figure 4 illustrates the superior quality of the novice-based sketch conditioning using our method against all the other baselines. KnobGen not only captures the spatial layout of the input sketch thanks to the CGC module but also extends beyond it by generating fine-grained details through the FGC module which ultimately produces a naturally appealing images. Whereas the baselines either rigidly conditions themselves on the imperfect input sketch or does not follow the spatial layout desired by the user.

| Models | CNet | T2I | UC | CNet++ | ADiff | KG-CN | KG-T2I |
|---|---|---|---|---|---|---|---|
| CLIP ↑ | 0.3214 | 0.3152 | 0.3210 | 0.3204 | 0.2988 | **0.3353** | 0.3271 |
| FID ↓ | 106.25 | 109.75 | 95.30 | 99.51 | 119.01 | **93.87** | 98.41 |
| Aesthetic ↑ | 0.5182 | 0.5093 | 0.5133 | 0.5253 | 0.4751 | **0.5349** | 0.5208 |

Table 1. Model comparison on CLIP, FID, and Aesthetic scores. Models include ControlNet (CNet), T2I-Adapter (T2I), UniControl (UC), ControlNet++ (CNet++), AnimateDiff (ADiff), and KnobGen variants (KG-CN, KG-T2I) with ControlNet and T2I-Adapter as Fine-Grained Controllers, respectively. KnobGen variants consistently outperform other models.

## 4.3. Quantitative Results

Table 1 provides a quantitative comparison between state-of-the-art DM models and KnobGen over 600 sketch images. We evaluated our model with two different FGC module plugins, that is, ControlNet and T2I-Adapter. We call our KnobGen whose FGC module is ControlNet KG-CN and with the T2I-Adapter KG-T2I. We measure performance using the CLIP score (prompt-image alignment), Fréchet Inception Distance (FID) and Aesthetic score (for more information, see Appendix 6). KG-CN achieves the highest CLIP score of 0.3353, surpassing the best baseline of 0.3214. KG-CN also gives the lowest FID score (93.87) and the highest aesthetic score (0.5349), demonstrating superior image quality and realism. We use a stratified sampling method based on pixel count to evaluate professional and amateur sketches, ensuring robustness across varying complexity levels. Our results demonstrate KnobGen's effectiveness in generating high-quality images, regardless of input skill level.

## 4.4. Ablation Study

One of the key innovations in our methodology is the introduction of the Modulator, a mechanism designed to enhance the training process of our proposed CGC module. We conducted an experiment where we trained two versions of KnobGen with Modulator and without it. To assess the effectiveness of the Modulator at the inference, we excluded the FGC module after 20 denoising steps in the image generation process ($S = 50$, and $\gamma = 20$, please refer to section 3.4). Excluding the FGC module imposes the conditioning of DM to be done by the CGC module. This experimental configuration demonstrates the power of our CGC module.

Figure 6. presents the results of these experiments, showcasing images generated with and without the Modulator. The comparative analysis reveals that the model trained with the Modulator exhibits a significantly enhanced ability to integrate *sketch-based coarse-grained guidance* into the image generation process. This indicates that the Modulator not only improves the model's overall performance but also ensures that the CGC's influence is effectively optimized during training, resulting in controlled image synthesis.

# 5. Conclusion

In this paper, we presented KnobGen, a dual-pathway framework designed to address the limitations of existing sketch-based diffusion models by providing flexible control over both fine-grained and coarse-grained features. Unlike previous methods that focus on detailed precision or broad abstraction, KnobGen leverages both pathways to achieve a balanced integration of high-level semantic understanding and low-level visual details. Our novel modulator dynamically governs the interaction between these pathways during training, preventing over-reliance on fine-grained information and ensuring that coarse-grained features are well-established. Additionally, our inference knob mechanism offers user-friendly control over the level of professionalism in the final generated image, allowing the model to adapt to a spectrum of sketching abilities—from amateur to professional. By incorporating these mechanisms, KnobGen effectively bridges the gap between user's input and model robustness. Our approach sets a new standard for sketch-based image generation, balancing precision and abstraction in a unified, adaptable framework.

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
