# OpenReview forum: "KnobGen: Controlling the Sophistication of Artwork in Sketch-Based Diffusion Models"
_thecvf.com/CVPR/2025/Workshop/CVEU — CVPR 2025_

### Official Review · Reviewer_GxpK · 2025-03-12

**Rating:** 3
**Confidence:** 4

**Review:**

==== Summary ====

This paper aims to provide an easy-to-use sketch-based generation model for both expert and novice users. At the core of the proposed KnobGen framework is a dual-branch conditioning mechanism, which balances the strength between micro, detailed and macro, abstract control. The micro branch adopts previous designs e.g. IP-Adapter and ControlNet. The macro control is a cross-attention layer fusing sketch input and text input. In addition, The author also designs two modulation mechanisms that alter the strength of the micro branch during both training and inference.

==== Strengths ====

- When using both image and text conditioning, how to balance the two conditions is a long-standing problem. The macro controller and modulators designed in this paper is an interesting solution to this problem.
- I like the visuals, especially Fig. 4, 6, 7 in the paper. They help explain the motivation and the effect of each module well.
- The performance gain in quantitative evaluation (Tab. 1) seems solid.

==== Weaknesses ====

- The inference time modulation mechanism is not new. It is known that we can apply guidance only on a subset of denoising timesteps to improve results. See, e.g., MultiDiffusion [1].
- I don't really understand the Coarse-Grained Controlled Module presented in Fig. 5 B. The text says this module fuses CLIP image and text features via cross-attention. However, the figure only shows some Conv and Transformer layers. Are these Transformer layers "cross-attention"? Also, how do you apply Conv to text embeddings that are 1D features? This doesn't make much sense to me.
- The paper has many references to the Appendix, yet the Appendix is not presented in the submitted PDF. Several important details are missing, such as the evaluation datasets & settings, more qualitative results, and user study results. If the authors can provide the Appendix (and assuming the Appendix is correct), this paper can get a higher score of Weak Accept. However, without these results, I can only recommend Borderline.

**Minor typos / writing issues:**
- line 276, "Clip" --> "CLIP"
- line 299, section name, "training" --> "Training"
- line 306, "overfits low-level features" --> "overfits to low-level features"
- The Conclusion section seems to have an inconsistent line spacing which are smaller than other sections

[1] Bar-Tal, Omer, et al. "MultiDiffusion: Fusing Diffusion Paths for Controlled Image Generation." International Conference on Machine Learning. PMLR, 2023.

---

### Official Review · Reviewer_VG7C · 2025-03-14
**Reviews for KnobGen**

**Rating:** 4
**Confidence:** 5

**Review:**

The paper presents a highly practical and meaningful problem: considering users' varying sketching abilities and the fidelity of image diffusion models in generating images from sketches of different levels.

The proposed method is direct, reasonable, and effective. The paper clearly articulates the authors' analysis of the problem, previous approaches, and motivation, and it provides a well-structured and detailed description of the proposed method.

The paper also introduces an interesting HCI design, allowing users to control the degree of explicit conditioning. This design is a clever and insightful approach because strictly enforcing diffusion models to adhere to explicit conditions can compromise the model's creativity. Providing users with a flexible control mechanism to adjust the fidelity of the generation is an innovative and engaging idea.

However, there are several issues that may have contributed to the paper not being accepted at a major conference. The authors may consider revising the manuscript accordingly:

1. Lack of in-depth analysis of structured information: Sketches can be viewed as a structured explicit condition, which is highly related to the learning patterns of diffusion models at different time steps. For example, earlier timesteps in the diffusion process tend to focus on learning the image layout, while later timesteps refine textures. However, the proposed method does not analyze these temporal differences or characteristics and instead applies conditional embeddings uniformly across all timesteps.

2. Limited experimental coverage: While I strongly agree with the problem presented in the paper, the choice of examples should highlight the shortcomings of previous methods more explicitly to better demonstrate the effectiveness of the proposed approach. Additionally, the comparison with baselines is limited to novice sketches, without evaluating the method on more complex sketches. Furthermore, internal feature visualizations should be included to illustrate both the weaknesses of prior methods and the strengths of the proposed approach.

3. Overly narrow focus on explicit information analysis and learning: Sketches can be considered a sparse form of conditioning, but there exist other, denser conditioning signals, such as flow and depth. These signals reflect different levels of user understanding of the drawing process. How such conditioning signals can be integrated into the diffusion training and inference processes—particularly considering the varying characteristics of different timesteps—warrants further exploration.

Addressing these points would significantly strengthen the paper and improve its impact in the field.

A little suggestion: The teaser that only occupies one column in the double-column format generally does not appear before the abstract. It is recommended that fig1 on the first page be moved to the right.

---

### Official Review · Reviewer_xT28 · 2025-03-25
**Weak knob value controls; missing dataset descriptions**

**Rating:** 2
**Confidence:** 3

**Review:**

Weaknesses:
1. The knob values are not strictly monocular with respect to conformity to the input sketch guidance. For example, in Figure 2, even when the knob values are close to 1, the wine bottle does not align with the input sketch. Same case for the second-row example in Figure 7. This undermines the claim of controllability using knob values.
2. There are no descriptions of the newly collected dataset or any details on the training datasets used.
3. The appendix is not submitted, despite being referred to in the main text.
4. Ablation studies do not come with quantitative evaluation. With the small amount of examples shown, it's hard for readers to derive reliable conclusions on the effect of the modulator. The hyperparmeters $\gamma$ and $S$ should also be ablated.

---

### Decision · Program_Chairs · 2025-03-25

**Decision:**

Accept

**Comment:**

The paper proposes KnobGen, a sketch-based generative model using a dual-branch conditioning mechanism for balancing detailed and abstract user control. Reviewers praised its practical relevance, clear motivation, and innovative control design, but expressed concerns about unclear explanations, missing dataset/training details, and insufficient quantitative evaluation.

Overall, the paper is accepted. Authors should clearly address reviewer feedback in the camera-ready version.